



**Thirteen-years of observations on primary sugars and sugar alcohols over remote**
**Chichijima Island in the western North Pacific**
Santosh Kumar Verma[1,2], Kimitaka Kawamura[1,3*], Jing Chen[1,4], and Pingqing Fu[1,5]
[1] Institute of Low Temperature Science, Hokkaido University, N19, W8, Kita-ku, Sapporo 060-
0819, Japan
[2] Permanent address: State Forensic Science Laboratory, Home (Police) Department,
Government of Chhattisgarh, Raipur 49001, India
[3] Now at Chubu Institute for Advanced Studies, Chubu University, Kasugai 487-8501, Japan
[4] Now at Institute of Geographic Sciences and Source Research, Chinese Academy of Sciences,
Beijing 100101, China
[5] Now at LAPC, Institute of Atmospheric Physics, Chinese Academy of Sciences, Beijing
100029, China
*Corresponding author: kkawamura@isc.chubu.ac.jp



**Abstract**. In order to understand the atmospheric transport of bioaerosols, we conducted long-
term observations of primary sugars and sugar alcohols over remote Chichijima Island in the
western North Pacific from 2001 to 2013. Our results showed that concentrations of total sugar
compounds for 13 years ranged from 1.2 to 310 ng m$^{-3}$ (average, 46±49 ng m$^{-3}$). We found that
atmospheric circulations significantly affect the seasonal variations of bioaerosol distributions
over the western North Pacific. The primary sugars (glucose and fructose) maximized in
summer, possibly due to a decreased transport of Asian aerosols and increased local emission
of vegetation products from the vascular plants in Chichijima. We also found higher
concentrations of sugar components (arabitol, mannitol and trehalose) in more recent years
during summer/autumn, suggesting an enhanced emission of fungal and microbial species over
the island. Sucrose peaked in late winter to early spring, indicating a springtime pollen
contribution by long-range atmospheric transport, while elevated concentrations of sucrose in
early summer could be assumed to be long-range transport of soil dust from Southeast Asia to
Chichijima. Positive matrix factorization analyses suggested the locally emitted sugar
compounds as well as long-range transported air borne pollen grains, microbes and fungal
spores are the major contributors to total sugar compounds in the Chichijima aerosols.
Backward air mass trajectories support the atmospheric transport of continental aerosols from
the Asian continent during winter/spring over Chichijima.
**Keywords:** Sugar compounds, fungal and microbial tracer, pollen tracer, bioaerosols, the
western North Pacific.





## 1 Introduction

East Asia has experienced rapid economic developments and population growth since last
several decades (Elliot et al., 1997; Jaffe et al., 1999, 2003), whose activities emit organic and
bioaerosols into the atmosphere (Xu et al., 2011). The atmospheric particles are transported to
downwind region in the Pacific, associated with Asian desert dust from the Taklamakan and
Gobi Deserts, and Loess plateau (Duce et al., 1980; Iwasaka et al., 1983; Jaffe et al., 1997;
Prospero and Savoie, 1989; Talbot et al., 1997). The transported dust contains bacterial cells,
fungal spores, and microbial cells, which fall out over the Pacific and remote islands in the
Pacific Ocean (Lacey and West, 2006; Mims and Mims, 2003). The microbes associated with
bioaerosols significantly affect the natural environment of marine and land ecosystem in
downwind regions (Graffin et al., 2003, 2007; Prospero et al., 2005). Long-range atmospheric
transport plays a key role for the global distribution of microbes from source regions to
receptor site (Graffin et al., 2001). Fungi and bacteria are often attached to dust particles,
which can propagate diseases to human and plants (Brown and Hovmoller, 2002). Therefore,
the transported organic and bioaerosols have been the focus of extensive studies for the past
years (Yamaguchi et al., 2012).
Organic aerosols are composed of a complex mixture of different types of molecules, in
which water-soluble organic compounds (WSOCs) are enriched (Graham et al., 2002). WSOCs
play an important role in climate change and global radiative forcing by scattering or absorbing
light directly or indirectly (Fuzzi et al., 2007). They can act as cloud condensation nuclei
(CCN) (Kanakidou et al., 2005; Martin et al., 2010). Sugar compounds (SCs) contribute 13–
26% and 63% of total WSOCs identified in continental and marine aerosols, respectively
(Simoneit et al., 2004a). SCs are directly emitted from biological sources such as fungi, algae,
pollen, spores and bacteria (Carvalho et al., 2003; Wang et al., 2009) and transported long
distances in the atmosphere (Wang et al., 2011). They are also derived from suspended soil



particles and associated biota (Rogge et al., 2006; Simoneit et al., 2004b; Wang et al., 2009),
and biomass burning (Schmidl et al., 2008; Simoneit et al., 2002).

Primary sugars are emitted from biological sources (Medeiros et al., 2006). Glucose and

fructose are emitted from terrestrial plant fruits, pollen, and detritus of vascular plants (Cowie
and Hedges, 1984; Speranza et al., 1997). Previous studies reported that sucrose is dominant
sugar component in airborne pollen grains and plays a significant role in plant blossoming
activity (Bieleski, 1995; Fu et al., 2012; Pacini, 2000). Trehalose is emitted from fungal
metabolic activities and resuspension of soil particles and unpaved road dust (Rogge et al.,
2007; Simoneit et al., 2004). Sugar alcohols are also emitted form biological sources like fungi
and microbes via metabolic activities (Bauer et al., 2008). Sugar alcohols, i.e., arabitol and
mannitol, are tracers for fungal spores (Jia and Fraser; 2011; Yang et al., 2012).  Di Filippo et
al. (2013) reported that arabitol and mannitol are key sugar components in fungal spores.

Chichijima Island is located in the western North Pacific: an outflow region of Asian

dust (Mochida et al., 2003a). Thus, particular attention has been paid to atmospheric chemical
studies over Chichijima Island. It is one of the best remote islands to study a long-range
transport of Asian aerosols, because local pollutants in Chichijima are insignificant due to low
population density and no major industrial or anthropogenic activities (Chen et al., 2013).
Kawamura et al. (2003) reported that lower molecular weight fatty acids derived from marine
organisms showed higher concentrations in summer, while higher molecular weight fatty acids
($C_{21}$-$C_{34}$), n-alkanes ($C_{25}$-$C_{35}$), n-alcohols ($C_{20}$-$C_{34}$) and dicarboxylic acids ($C_{20}$-$C_{28}$) derived
from terrestrial higher plants and soil organic matter maximized in winter to spring. Seasonal
variations of low molecular weight dicarboxylic acids and levoglucosan (biomass burning
tracer) have been discussed in Chichijima aerosols by Mochida et al. (2003a) and Mochida et
al. (2010), respectively. Although seasonal variation of saccharides was reported in Chen et al.
(2013), the observation period is rather short. Therefore, long-term observations are needed to



obtain a data set of SCs to better discuss the characteristics, potential sources, and possible
effects of atmospheric transport over the western North Pacific.

Here, we report thirteen-year data set of SCs in remote Chichijima Island. The goal of

this study is to characterize seasonal and annual variations of SCs and specify their possible
source regions. We will also discuss a potential role of Asian dust to control the distributions
of bioaerosols over the western North Pacific. The outcomes of this study will improve our
understanding about a possible influence of long-range transport of bioaerosols from the
continent to the clean oceanic environment. We will compare the data set of SCs for the
periods 1990-1993, 2001-2003 and 2010-2013, which may provide imperative information
about decadal changes in the atmospheric conditions over Chichijima. Seasonal source
identifications by positive matrix factorization (PMF) analysis will also be discussed for the
measured SCs.

**2  Materials and methods**
**2.1  Sampling site and meteorological conditions**
The detailed information on the sampling site was reported in Kawamura et al. (2003) and
Chen et al. (2013). Briefly, Chichijima Island is located in the western North Pacific (27°04'N;
142°13'E), 1000 km south of Tokyo, Japan, and 2000 km east of the Asian continent (Figure 1).
Total area of the island is 24 km$^2$ with a population of 2000 (Verma et al., 2015). The climate
of Chichijima is classified as subtropical; it is warm to hot (temperature, 7.8-34.1 °C) and
humid (relative humidity, 66-88%) all year round.

Figure 2 shows monthly averaged variations in the meteorological parameters of

Chichijima during 2001-2013. It receives more precipitation in between April and July,
September and October during the thirteen-year periods. The sampling site is less influenced
by the East Asian monsoon to receive heavy rainfall compared to Northeast Asia. The climate
over Chichijima is strongly influenced by the seasonal changes in wind system. In



winter/spring, the westerly winds are dominant with the air masses being enriched with Asian
dust, industrial pollutants, biomass burning products, organic compounds and black carbon as
well as bioaerosols emitted from East Asia and Eurasia (Figure 3) (Seinfeld et al., 2004;
Simoneit et al., 2004b; Wang et al., 2009). Trade winds are dominant in summer/autumn,
which transport clean and pristine marine air masses from the central Pacific to Chichijima
(Kawamura et al., 2003, Mochida et al., 2010).
**2.2   Aerosol sampling and chemical analysis**
The details on aerosol sampling and chemical analysis are reported elsewhere (Chen et al.,
2013; Mochida et al., 2010). Briefly, total suspended particle (TSP) samples were collected at
the Ogasawara Downrange Station of the Japan Aerospace Exploration Agency (JAXA) in
Chichijima Island (254 m, above sea level, asl). The samples were collected on weekly basis
(January 2001 to November 2013) using a high volume air sampler (Kimoto AS-810A) at a
flow rate of 1.0 $m^3$ $min^{-1}$ and pre-combusted (450°C for 6 h) quartz fiber filters (20 x 25 cm,
Pallflex). Filter sample was placed in a pre-combusted glass jar with a Teflon-lined screw cap
and stored in a dark freezer room at -20 °C prior to analysis in order to inhibit fungal growth.
Due to the maintenance of the JAXA facility at sampling site, TSP samples were not collected
for November–December 2004 and March–August 2005.
Total 590 aerosol samples were analyzed to determine the primary sugars (xylose,
fructose, glucose, sucrose and trehalose) and sugar alcohols (erythritol, arabitol, mannitol and
inositol) during 2001 to 2013. An aliquot (21 $cm^2$) of the filters were extracted three times with
dichloromethane/methanol (2:1, v/v) mixture using ultrasonic agitation for 10 minutes. A
Pasture pipette packed with quartz wool was used to remove particles and filter debris in the
extracts. Filtrates were then concentrated using a rotary evaporator under vacuum and blown
down with a stream of pure nitrogen gas. The total extracts were derivatized using 60 µl of
N,O-bis-(trimethylsilyl)trifluoroacetamide (BSTFA) with 1% trimethylsilyl chloride in the
presence of 10 µl of pyridine in a sealed vial at 70 °C for 3 hours to convert hydroxyl groups to





corresponding trimethylsilyl (TMS) ethers. The derivatized fractions were diluted with n-
hexane containing internal standard of $C_{13}$ n-alkane (1.43 ng μl$^{-1}$), prior to injection to gas
chromatography-mass spectrometer (GC-MS).

Identification of SCs have been confirmed by the comparison of GC retention times and

mass spectra with those of authentic standards as well as literature and library data. SCs were
characterized by their common base peak at m/z 217 and 204 with specific fragment ions for
individual sugars, i.e., m/z = 307 (arabitol), 205 and 319 (mannitol), 205 (erythritol), 305 and
318 (inositol), 361 (sucrose and trehalose), 191 (glucose), and 437 (fructose). The selected ion
peak area and relative response factors determined by injection of authentic standards have
been used for the quantification of sugar compounds. Field blank filters were analyzed as a real
sample, but no target compounds were detected in the field blanks. The recoveries of the target
compounds were better than 90%. Therefore, the data reported here were not corrected for
recoveries. Analytical errors of SCs were generally <15% based on duplicate analysis. The
detection limits of primary sugars and sugar alcohols were 105-557 pg μl$^{-1}$, which corresponds
to ambient concentrations of 0.0015-0.0081 ng m$^{-3}$ under a typical sampling volume of 9000
m$^3$ (Zhu et al., 2015).

The derivatized fractions were introduced into GC-MS using an Agilent model 7890 GC

coupled to an Agilent model 5975 mass selective detector (MSD) operated in an electron
impact mode at 70 eV and scanned from 40 to 650 Dalton. The GC separation was carried out
on a DB-5MS fused silica capillary column (30 m long, 0.25 mm i.d., 0.25 μm film thickness),
with a temperature program of 50 °C for 2 min at a rate of 15 °C min$^{-1}$ from 50 to 120 °C, then
from 120 to 305 °C at a rate of 5 °C min$^{-1}$ with a final isotherm hold at 305 °C for 15 min. The
sample was injected on a splitless mode at an injector temperature of 280 °C. GC-MS data
were acquired and processed with the Agilent GC/MSD ChemStation software.
**2.3  Backward air mass trajectory analysis**



In order to identify the source regions of sugar compounds in Chichijima aerosols, ten-day
backward trajectories were calculated at 00:00 UTC of each sampling period for thirteen-years
using the NOAA Hybrid Single-Particle Lagrangian Integrated Trajectory
(http://ready.arl.noaa.gov/HYSPLIT.php) (Figure 3). The starting height of the trajectories
presented in this study is 500 m asl. We plotted thirteen-year trajectories for each sampling day
but there are no significant year-to-year changes in the atmospheric circulations. Therefore, we
presented seasonal trajectories for recent year (December, 2011 to November, 2012) in Figure
3 to understand the seasonal aerosol mass transport from the source regions to Chichijima
Island. Backward trajectories significantly supported a long-range transport of air mass under
the influence of existing meteorological parameters (Figure 3). The trajectories clearly show
the influences of continental air masses during mid-autumn to mid-spring and of marine air
masses during mid-spring to mid-autumn.
**2.4  Positive matrix factorization (PMF) analysis**

Positive matrix factorization (PMF 3.0, Environmental Protection Agency, USA) has

been used as a powerful statistical tool that may resolve potential sources contributing to
atmospheric levels of particle (as presented by %) when appropriate source profiles are not
available (Paatero and Tapper, 1994). At the beginning PMF has been used in precipitation
study (Juntto and Paatero, 1994) as well as air pollution and source apportionment studies
(Polissar et al., 1999). Recently, it is widely using for the air quality and source apportionment
(Xie and Berkowitz, 2006). In addition, PMF has been applied to the wastewater
(Soonthornnonda and Christensen, 2008), lakes sediments (Bzdusek et al., 2006) and soils (Lu
et al., 2008). One of the main features of PMF results is their quantitative nature; it is possible
to obtain the composition of the sources determined by the model.

PMF uses uncertainties for each of the measured concentration ($x_{ij}$). PMF minimize the

residual sum of squares (Q) defined by equation:





$$Q = \sum_{i=1}^{n} \sum_{j=1}^{m} \left(\frac{e_{ij}}{s_{ij}}\right)^2$$


Where the $j$ is species in the $i^{th}$ sample, $e_{ij}$ is portion of the measurements. In PMF, only
measured concentrations ($x_{ij}$) are known and the goal is to estimate the contributions and the
fractions. The uncertainties were computed from the measurement errors by equation:

$$s_{ij} = \sigma_{ij} + C_3 \max\left(|x_{ij}|, |y_{ij}|\right)$$


Where $y_{ij}$ is the calculated value for $x_{ij}$, $\sigma_{ij}$ is the measurement or estimated error, and $C_3$ is a
dimensionless constant value. The estimation of the measurement errors of size distribution
data were based on the combination of size bins (Zhou et al., 2004). $C_3$ is used as the
estimation of the relative uncertainties of large values (Norris et al., 2008). Using too few
factors will combine sources of different nature together and using too many factors will make
a real factor further dissociate into two or more non-existing factors. $F_{peak}$ is a parameter in
PMF for controlling rotations (Paatero et al., 2002). When the $F_{peak}$ value is positive, the
following additional term is included in the object function Q:

$$Q^P = \beta^2 \left(\sum_{K=1}^{P} \sum_{J=1}^{N} f_{kj}\right)^2$$


where, $\beta^2$ corresponds to the $F_{peak}$ value.
PMF analysis was performed for quantitative estimation of sources for the collected
samples using tracer compounds for primary sugars, sugar alcohols, and anhydrosugars. Based
on given understanding of sugar sources, 4-7 factors were examined and total five interpretable
factors were characterized by the enrichment of each tracer compound, which reproduced more
than 94% of SCs. Minimal robust and true Q values of the base run were 3001 and 3413,
respectively. Concentrations and percentage of tracers in each factor of bootstrap run were
close of those of base run results. The Q values and factor profiles of $F_{peak}$ rotation runs showed
no significant changes compared with base run, indicating stable PMF results.





In winter/spring, Chichijima Island receives air masses enriched with anthropogenic
aerosols from the Asian continent by strong westerly winds, whereas during summer/autumn it
receives clean air masses from the Pacific Ocean under the influences of trade winds. The
seasonal changes in the atmospheric circulation over Chichijima may have a significant
influence on the seasonal distributions of SCs. Therefore, we performed the seasonal PMF
analysis on the thirteen-year sugar data set to better understand the seasonal source profile of
individual sugar component. For seasonal PMF analysis, 3-5 factors were examined and 4
factors were determined for each season. We included the data set of anhydrosugars from
Verma et al. (2015) for PMF analysis.

**3. Results and discussion**
**3.1 Ambient concentrations of sugar compounds**
Temporal variations of primary sugars and sugar alcohols are shown in Figure 4. Nine sugar
compounds (SCs) including five primary sugars and four sugar alcohols were detected in the
aerosol samples collected from Chichijima Island. The concentrations of total SCs varied from
1.23 to 339 ng m$^{-3}$ (average, 46.7±49.5 ng m$^{-3}$) during 2001 to 2013 (Table 1). Concentrations
of primary sugars and sugar alcohols were in the range of 0.28 to 176 ng m$^{-3}$ (23.3±25.7 ng m$^{-3}$
) and 0.37 to 231 ng m$^{-3}$ (23.4±30.8 ng m$^{-3}$), respectively. Average concentration of primary
sugars in Chichijima aerosols is several times lower than that of primary sugars (62.0±54.9 ng
m$^{-3}$) reported from Cape Hedo, Okinawa, Japan (Zhu et al., 2015) while that of sugar alcohols
is equivalent to or little lower than that from Cape Hedo (29.5±35.5 ng m$^{-3}$).
Interestingly, primary sugars (49.9%) and sugar alcohols (50.1%) were found to
contribute almost equal to total SCs during the entire study period. Mannitol (26.7%) and
arabitol (21.4%) were the main contributors to total SCs followed by glucose (16.7%), sucrose
(13.6%), fructose (10.2%), and trehalose (9.2%). Erythritol (1.6%), inositol (0.3%), and xylose
(0.3%) were also present in the aerosols at lower concentration levels. Temporal plots of





individual sugars clearly indicate a large variation of SCs (Figure 4). This large variation in the
concentrations of SCs might be involved with seasonal changes in the atmospheric circulations
in the western North Pacific (Kawamura et al., 2003).
**3.1.1 Concentrations of primary sugars in total SCs**
Glucose is the dominant sugar species among the primary sugars with concentration range of
0.05 to 64.3 ng m$^{-3}$ (average, 7.79±8.80 ng m$^{-3}$). Similarly, a wide concentration range of
fructose (0.03-115 ng m$^{-3}$; 4.69±8.04 ng m$^{-3}$) was also observed in Chichijima aerosols.
Thirteen-year mean concentrations of glucose and fructose were observed to be lower than
those (27.2 ng m$^{-3}$ and 16.4 ng m$^{-3}$, respectively) reported for the aerosol samples (TSP) from
Cape Hedo, Okinawa, Japan (Zhu et al., 2015). Glucose and fructose significantly contribute to
total primary sugars (33.5% and 20.17%, respectively) in Chichijima aerosols. The primary
sugars are abundant in the fragments of vascular plants in vegetated and forest areas (Medeiros
et al., 2006). Pacini et al. (2000) reported that primary sugars are synthesized in leaves during
photosynthesis and stored in root, stem, flower, pollen and fruit of growing plants. The nectars
and fruits of tropical and subtropical plants also contain glucose and fructose abundantly
(Backer et al., 1998). Graham et al. (2002) reported significant amounts of glucose and
fructose in pollen, fern spores, and insects in aerosol samples collected from the Amazon forest.
Chichijima Island is covered with endemic and vascular plants, which may emit glucose and
fructose. Moreover, different sources such as soil dust (Rogge et al., 2007; Simoneit et al.,
2004), lichens (Dahlman et al., 2003) and biomass burning (Medeiros et al., 2006; Nolte et al.,
2001) have also been reported as dominate sources for glucose and fructose.

Among all the SCs detected in the Chichijima aerosols, sucrose is the second most

abundant sugar species (0.002-100 ng m$^{-3}$; 6.43±12.9 ng m$^{-3}$), accounting for 27.3% of total
primary sugars. The average sucrose concentration observed in Chichijima is twice lower than
that (13.2 ng m$^{-3}$) from Cape Hedo, Okinawa, Japan (Zhu et al., 2015). Sucrose is synthesized
in plant leaves and circulated by phloem to different plant sections, which is accumulated in





root cells as well as developing flower buds (Bieleski, 1995; Jia et al., 2010). Sucrose is a
dominant component in airborne pollen grains derived from flowering plants (Bieleski, 1995;
Pacini, 2000). Simoneit et al. (2004a and 2004b) reported the presence of sucrose in surface
soil and paved road dust. Sucrose was also observed in dry plant materials during harvesting
period (Ma et al., 2009).

Thirteen-year mean concentration of trehalose ranged from 0.01 to 70.2 ng m$^{-3}$

(4.30±7.28 ng m$^{-3}$), whose average concentration accounts for 18.4% of total primary sugars
detected in Chichijima aerosols for 13 years. Microbes (bacterial cell), fungal spores, yeast,
algae, invertebrates, suspended soil dust, as well as plant species, contribute significantly to
trehalose in the atmosphere (Elbein, 1974; Graham et al., 2003; Medeiros et al., 2006; Rogge et
al., 2007; Simoneit et al., 2004; Wiemken, 1990). Xylose is a less abundant primary sugar,
accounting for 0.60% of total primary sugars observed in Chichijima aerosols. The
concentration range of xylose was 0.001-1.35 ng m$^{-3}$ (0.14±0.18 ng m$^{-3}$) during sampling
period of thirteen years. Biomass burning activities emit xylose to the atmosphere. Cowie and
Hedges (1984) reported that xylose produced by angiosperm and gymnosperm plants,
phytoplankton, as well as groups of microorganisms. Simoneit et al. (2004a) have reported
xylose in soil dust from various locations in the United States and Japan. Wan and Yu (2007)
also observed xylose in soils and associated micro biota.
**3.1.2  Concentrations of sugar alcohols in total SCs**
Thirteen-year mean concentrations of arabitol and mannitol were found to be 9.99±13.6 ng m$^{-3}$
and  12.5±17.5 ng m$^{-3}$, which contribute to 42.7% and 53.3% of total sugar alcohols,
respectively. The concentration ranges of arabitol (0.04–106 ng m$^{-3}$) and mannitol (0.10–118
ng m$^{-3}$) are comparable to those from the Mediterranean region, Israel (arabitol, 1.85–58.3 ng
m$^{-3}$ and mannitol, 5.57–138 ng m$^{-3}$) (Burshtein et al., 2011). Yttri et al. (2007) also reported
that arabitol and mannitol were main contributors of sugar alcohols in aerosol samples
collected from the different background sites in Norway. Sugar alcohols (arabitol, mannitol)



can be used as tracers for various fungal and algal species (Bauer et al., 2008a,b; Pashanska et
al., 2002; Zhang et al., 2010). Loos et al. (1994) discussed arabitol and mannitol as potential
sources of bacteria and other microbes. High levels of detritus from the spring bloom and
autumn decomposition have been reported as significant sources for arabitol and mannitol in
the vegetated region (Burshtein et al., 2011; Pashynska et al., 2002). Good positive correlations
of arabitol (r = 0.63) and mannitol (r = 0.72) with glucose indicate a vegetation contribution of
both sugar alcohols in Chichijima aerosols. Erythritol and inositol are less abundant sugar
species, accounting for 3.29% and 0.66% of total sugar alcohols. Their concentration ranges
are 0.01-8.32 ng m$^{-3}$ and 0.01-1.81 ng m$^{-3}$, respectively. Significant positive correlations of
both sugar species with arabitol and mannitol indicate similar sources for these SCs in
Chichijima aerosols (Table 2).

**3.2  Seasonal variations of total sugar compounds**

Seasonal concentration range, mean and median values of individual sugars during the

study periods of thirteen-years are presented in Table 1. The concentrations of individual
sugars were extensively fluctuated from season to season in aerosol samples collected at
Chichijima (Figures 4 and 5a). The seasonally averaged concentrations of total SCs are higher
in summer (71.5±70.9 ng m$^{-3}$) and autumn (57.0±64.2 ng m$^{-3}$) than spring (39.8±67.6 ng m$^{-3}$)
and winter (18.2±34.0 ng m$^{-3}$) over Chichijima Island. Zhu et al. (2015) measured sugar
components in aerosol samples collected from Cape Hedo, Okinawa, Japan and reported 2 to 3
times higher concentrations in summer (136 ng m$^{-3}$) and spring (133 ng m$^{-3}$) than autumn (86
ng m$^{-3}$) and winter (40 ng m$^{-3}$), whose seasonal trends are similar to Chichijima. Wan and Wu
(2007) reported different seasonal variations with the highest concentration in autumn (375 ng
m$^{-3}$), followed by winter (292 ng m$^{-3}$) and spring (84 ng m$^{-3}$) for the continental urban aerosols
collected from Hong Kong. These concentrations in Hong Kong are 16 and 6 times higher than
those of the remote Chichijima samples for winter and autumn, respectively. Interestingly, the
different seasonal trends between the continental urban sites and two islands in the western





North Pacific may be associated with different sources and transport pathways between the
urban and marine environments.

### 3.2.1 Seasonal variations of primary sugars

Glucose maximized in summer ($11.0\pm9.02$ ng m$^{-3}$) followed by autumn ($9.25\pm8.63$ ng m$^{-3}$),
spring ($7.68\pm10.3$ ng m$^{-3}$) and winter ($3.11\pm3.53$ ng m$^{-3}$) (Table 1 and Figure 5h). Glucose is
the most abundant primary sugar in Chichijima aerosols. In winter/spring, Chichijima is
influenced by strong westerly winds that deliver the air masses the Asian continent including
Mongolia, Russian Far East and north China, where vegetation is active. Consequently,
declined concentration of glucose in winter means the depressed emission of continental
bioaerosols from Asia in spite of a long-range transport of Asian dusts due to strong westerly.
The local vegetation (vascular plants) in Chichijima Island might be responsible to enhanced
glucose during growing season (spring and summer) and decaying periods of plant leaves
(autumn). Seasonal PMF analysis also supports dominant sources of vegetation for glucose
among four factors, which contributed >75% for mixed factor in summer (Figure 6c), >80%
for fungal and vegetation factor in autumn (Figure 6d), and >75% for vegetation factor in
spring (Figure 6b).

Fructose shows the highest concentrations in summer ($7.25\pm7.63$ ng m$^{-3}$) followed by

spring ($4.51\pm9.21$ ng m$^{-3}$), autumn ($3.70\pm2.68$ ng m$^{-3}$) and winter ($3.36\pm10.2$ ng m$^{-3}$). As
shown in Table 2, a significant correlation (r=0.57) was obtained between glucose and fructose.
Burshtein et al. (2011) reported similar correlations for both sugar species, suggesting an
identical input of glucose and fructose from the local vegetation in summer (Baker et al., 1998;
Pacini, 2000). Monthly mean concentrations of fructose show two prominent peaks in
February-March and June-July, the latter peak may be due to the local vegetation in Chichijima
(Figure 5g). The fructose peak in February-March may be influenced by air borne pollen grains
in the spring bloom of flowering plants. High concentration of fructose was observed in spring
followed by summer, indicating an input of this sugar compound from pollen grains (Fu et al.,





2012). The positive correlation of fructose with sucrose (pollen tracer) supports the similar
sources. Seasonal PMF analysis further supports the identical source for fructose and sucrose;
that is, among four factors, fructose contributes >70% and >60% for pollen factor in spring
(Figure 6b) and winter (Figure 6a), respectively.

Seasonal mean concentrations of trehalose showed a maximum in summer (7.06±8.49 ng

m$^{-3}$) followed by autumn (6.09±8.81 ng m$^{-3}$), spring (2.93±6.08 ng m$^{-3}$) and winter (1.03±1.26
ng m$^{-3}$) (Table 1). Monthly mean concentrations of SCs for 13 years show that concentrations
of trehalose are higher during June to October (Figure 5j). Ma et al. (2009) reported higher
concentrations of trehalose at the urban site of Guangzhou, China during summer and autumn.
Similarly, Wan and Wu (2007) reported a similar autumn maximum in Hong Kong. On the
other hand, different seasonal trends of trehalose were reported for the aerosol samples (TSP)
collected in USA (Medeiros et al., 2006), China (Wang et al., 2011), Australia (Hackl et al.,
2000), and Gosan, Jeju Island in the western North Pacific Rim (Fu et al., 2012). In the above
studies, highest concentrations of trehalose were reported in early spring due to the re-
suspension of soil particles during agricultural practice. Hackl et al. (2000) also obtained
abundant trehalose in spring, and proposed that trehalose can be used as a tracer of soil dust
emission to the atmosphere. However, we did not detect a spring peak of trehalose in
Chichijima aerosols, suggesting that soil dust contribution of trehalose over Chichijima is
insignificant via long-range atmospheric transport. Seasonal PMF analysis for autumn showed
that more than 85% of trehalose was contributed by microbial factor among four factors
(Figure 6d). An indirect contribution of trehalose from soil dust will be discussed later.

The seasonal mean concentrations of sucrose are almost equal during spring (8.80±18.0

ng m$^{-3}$), summer (7.31±11.5 ng m$^{-3}$) and winter (6.60±13.1 ng m$^{-3}$), except for autumn
(2.76±4.35 ng m$^{-3}$) (Table 1). The homogeneous seasonal distribution suggests multiple
sources of sucrose in Chichijima aerosols. Monthly mean concentrations of sucrose show two
peaks during February-March and June-July (Figure 5i). The March peak of sucrose was



reported in the forest area of Sapporo, Japan to be 3 to 7 times more abundant than other
months due to springtime pollen emissions (Miyazaki et al., 2012). Fu et al. (2012) analyzed
pollen samples from different plant species (white birch, Chinese willow, Peking willow) for
SCs and found the highest concentrations of sucrose followed by fructose and glucose in pollen.
The pollen emissions from developing buds of plants may be the reason for the increased
concentration of sucrose and fructose in February and March over Chichijima Island in the
western North Pacific. Seasonal PMF analysis shows that sucrose contributed 100% in spring
and >80% in winter, suggesting a significant pollen contribution for sucrose in those seasons
(Figures 6b and 6a).

However, the possibilities of pollen transport from East Asia to Chichijima cannot be

excluded because pollens can travel long distances with springtime high-speed winds by
westerlies (Rousseau et al., 2006). The pollen grains emitted from flowering boreal forest in
China, Mongolia, Siberia and Russian Far East, could significantly be delivered to the western
North Pacific during spring, which may contribute sucrose and fructose in Chichijima aerosols.
Recent studies have discussed a long-range transport of airborne pollen from North America to
Greenland in spring (Rousseau et al., 2008). Lorenzo et al. (2006) reported the long distance
transport of airborne allergenic pollen to central Italy. Makra et al. (2010) reported a long-
range transport of airborne pollen in three European cities by applying three-dimensional
clustering of backward trajectories. Several studies also discussed a long-range transport of
pollen to the remote arctic region (Andrews et al., 1980; Bourgeois et al., 2001; Campbell et al.,
1999; Hicks et al., 2001; Hjelmroos and Franzen, 1994; Rousseau et al., 2004).

These observations may support that westerly winds have delivered pollen grains from

the Asian continent including Mongolia, Siberia, and Russian Far East to Chichijima Island in
spring. Using a box model and typical settling velocity of pollens (3 cm/sec) with the grain size
of 30 μm in diameter (Sosnoskie et al., 2009), we estimated lifetime of pollen grains to be 9.3
hours in the atmospheric marine boundary layer (height of 1 km above the ocean surface). The



settling velocity of the pollens is ca. 20 times larger than that of typical marine aerosols (Slinn
and Slinn, 1980). Because pollen grain sizes range from 10 μm to 100 μm in diameter, the
lifetime of pollens may have a large uncertainty. If pollens could be largely transported in the
free troposphere (e.g., 5 km high) to the North Pacific from the Asian continent, then lifetime
of typical pollen grains would increase upto 2 days. These calculations for the lifetime of
pollen grains further support their long-range atmospheric transport from the Asian continent
over the western North Pacific. Based on backward air mass trajectories (Figure 3), we can
roughly estimate the transport time from East Asia to Chichijima site to be 2-4 days in winter
and spring. It is also of interest to note that pollens can rapture under condition of high relative
humidity (RH) (Hader et al., 2014; Miguel et al., 2006; Wright et al., 2014), which leads to
smaller particles with longer residence time in the atmosphere.

In addition, tilling process after wheat crop harvesting in farmland causes an enhanced

exposure of wheat root associated with soil particles into the atmosphere. China, India, and
USA are three largest countries for wheat production in the world. In China and India there are
two seasons (spring and winter) for wheat crops; winter wheat is harvested from mid-May to
mid-July. During those periods (early summer), Chichijima Island is highly influenced by trade
winds (Figure 3). However, trajectories clearly show the occasional air mass transport during
summer from the Southeast Asia to Chichijima (Pavuluri et al., 2010). PMF results of sucrose
for summer (Figure 6c) and autumn (Figure 6d) account for >85% and >90%, respectively, for
soil dust factor among four source factors, suggesting an additional source of soil dust for
sucrose in Chichijima (Simoneit et al., 2004). The elevated sucrose concentrations in June and
July (summer; non flowering seasons) suggest the transport of sucrose associated with soil
particles under the influence of occasional air mass transport in summer from Southeast Asia
(Figure 3).

Xylose was found as the least abundant sugar compound in the aerosol samples. The

maximum concentration of xylose (1.35 ng m$^{-3}$) was found in summer whereas minimum



(0.001 ng m⁻³) in spring (Table 1). Summer mean concentration (0.18±0.26 ng m⁻³) was highest
(Table 1). The PMF analyses showed that xylose contributed >75% for BB factor in winter
(Figure 6a) and >70% in autumn (Figure 6d) for microbial factor. These results suggest
different sources and seasons for xylose; i.e., biomass burning in winter (Sullivian et al., 2011)
and groups of microorganisms in summer (Cowie and Hedges, 1984).
**3.2.2 Seasonal variations of sugar alcohols**
The seasonal mean concentrations of arabitol and mannitol are higher in summer/autumn than
spring/winter (Table 1). The concentrations of arabitol are equally distributed between summer
(15.1±12.9 ng m⁻³) and autumn (15.8±18.3 ng m⁻³) with lower levels in spring (7.13±9.50 ng
m⁻³) and winter (1.73±2.60 ng m⁻³). Mannitol maximized in summer (21.7±19.7 ng m⁻³)
followed by autumn (18.2±19.9 ng m⁻³), spring (7.95±13.8 ng m⁻³) and winter (1.89±2.81 ng
m⁻³). Arabitol and mannitol strongly co-varied throughout the study period. As depicted in
thirteen-year monthly mean concentrations of total SCs (Figure 5b,c), we found elevated
concentrations of sugar alcohols from May to October. Similar seasonal trends were reported
for the aerosol samples collected from Gosan, Jeju Island in the western North Pacific Rim (Fu
et al. 2012) and urban aerosol samples from Ghent, Belgium (Pashynska et al., 2002). In above
studies, higher relative abundances of arabitol and mannitol in total sugar alcohols were
reported during late summer to autumn. The higher concentration of arabitol in autumn was
also reported for aerosol samples from the Mediterranean region in Israel (Burshtein et al.,
2011). Erythritol and inositol showed the similar seasonal trend, but their concentrations are
lower than the former two sugar species.
Sugar alcohols are emitted to the atmosphere from a variety of bacteria, few green algal
lichens and fungi (Dahlman et al., 2003; Filippo et al., 2013). Arabitol and mannitol are
abundant in fungal spores (Lewis and Smith, 1967; Yttri et al., 2007). Arabitol (r = 0.73) and
mannitol (r = 0.80) showed a strong co-variance with trehalose, suggesting identical sources of
sugar species in Chichijima. The PMF analysis showed that fungal factor and mixed factor



(fungal, vegetation, and microbial) accounted for 25% and 54.2% of total SCs observed in
summer, respectively (Figure 6c). In autumn fungal and vegetation factor contributed 71% of
total SCs detected in Chichijima aerosols (Figure 6c). In winter (Figure 6a) and spring (Figure
6b) fungal and vegetation factor and mixed factor account for 31.2% and 37.2% of total SCs,
respectively. This is reasonable because fungal and microbial activities are lower during
winter/spring as compared to summer/autumn. The meteorological factors such as RH and
temperature significantly affect fungal and bacterial activity (Kim and Xiao, 2005; Malik and
Singh, 2004). Higher RH and temperature are crucial in increasing fungal and bacterial growth
(Sharma and Razak, 2003). Their maximum growth was observed under the condition of 92-
100% RH (Ibrahim et al., 2011). Higher concentrations of arabitol and mannitol in summer and
autumn may be due to the increased fungal and bacterial activities in Chichijima Island in the
western North Pacific.

Several studies have described the occurrence of fungi in marine environment (Jones,

1976; Kohlmeyer and Kohlmeyer, 1991; Moss, 1986). The fungal species eject spores from
hard materials like coral and sand grains. Some fungi also eject spores from woods associated
with sand in summer/autumn when higher ambient temperature and RH are available (Jones
and Mitchell, 1996). Marine fungal growths are observed on several mediums of substrates
such as wood, sediments, muds, soil, sand, algae, corals, decaying leaves of mangroves and
living animals in marine environment (Bremer, 1995; Kohlmeyer and Kohlmeyer, 1979;
Nagakiri et al., 1996). However, the above studies have claimed to the occurrence and growth
of marine fungi on several mediums of substrates but still the knowledge of the role of marine
fungi in sediments and decaying dead animals are insufficient due to the lack of appropriate
data set. It is still unclear if these fungi are active in sediments or if inactive spores are isolated
(Hyde et al., 1998). Therefore, due to the inadequate data set, we doubt the marine contribution
of sugar alcohols (arabitol, mannitol) in Chichijima aerosols.



Thirteen-year monthly mean concentrations of SCs clearly show slightly decreased
concentrations of arabitol, mannitol, and erythritol in July and August; a similar trend was
observed for trehalose (Figure 5b,c,e,j). These sugar compounds are derived from the microbial
activities in source regions. The thirteen-year precipitation record over Chichijima Island
shows that precipitations were lowered in July and August (Figure 2). The lower precipitation
amount decreases the RH (Figure 2) and thus depresses the fungal and microbial activities. The
lower precipitation also suppresses the moisture contents in the surface soil of Chichijima,
which should cause a significant decline of local fungal and other microbial activities on the
ground of Chichijima Island. Decreased precipitation might be a possible reason for the lower
concentrations of arabitol, mannitol, and trehalose in July and August.
**3.3  Annual variations and decadal comparisons of SCs**
The annual variations in the concentrations of primary sugars and sugar alcohols are shown in
Figure 7a. The annual mean concentrations of total SCs varied randomly during 2001 to 2013.
As shown in Figure 7 (i, j, f and d), concentrations of sucrose, trehalose, xylose, and inositol
increase from 2001 to 2013 in Chichijima aerosols. Similarly, arabitol (Figure 7b), glucose
(Figure 7h), and fructose (Figure 7g) show clear increasing trends from 2006 to 2013 whereas
mannitol (Figure 7c) and erythritol (Figure 7e) show a random trend. Thirteen-year data set
provides significant information regarding the decadal variation in the atmospheric conditions
over Chichijima Island in the western North Pacific. Here, we compare the data set of SCs for
the periods of 1990-1993 (Period, P-I) with the current observations from 2001 to 2003 (P-II)
and 2010 to 2013 (P-III) (Table 3) (Figures 8a, b, c).
The comparison for three periods indicates that concentrations of anhydrosugars are
highest in winter followed by autumn. Their concentrations significantly increased from P-I to
P-II/P-III (Figure 8a). The detailed discussions on anhydrosugars were reported in Verma et al.
(2015). Here, we refer the data set of anhydrosugars for the decadal comparison with SCs.
Interestingly, biomass-burning tracers (BB tracers; levoglucosan, mannosan and galactosan)





showed a significant difference in the decadal trends among three periods (i.e., P-I, P-II, and P-
III) during winter and autumn. In winter, BB tracers showed an increasing trend from P-I to P-
III (Figure 8a). Biomass burning is common in winter for house heating (Simoneit et al.,
20004a), thus it is obvious that the lower ambient temperature is the more biomass burning
activities are. Westerly winds abundantly transport biomass-burning products over Chichijima
Island in the western North Pacific with air masses derived from East Asia, Siberia, Mongolia,
and Russian Far East during winter (Bendle et al., 2007; Simoneit and Elias, 2010; Verma et al.,
2015). In contrast, BB-tracers in autumn show an opposite trend (i.e., higher concentrations in
P-I followed by P-II and P-III) compared to those in winter (Figure 8a).

The difference in the concentrations of anhydrosugars in winter and autumn during P-I is

insignificant while the concentrations are 3 and 5 times higher in winter than autumn for P-II
and P-III, respectively. This seasonal shifting in the concentrations of anhydrosugars may be
attributable to the changes in the strength of both westerly and trade wind systems from mid
autumn to early winter among three periods (Chen et al., 2013). In contrast, the concentrations
of primary sugars were 2 to 7 times higher during P-II and P-III than P-I period in summer and
autumn (Figure 8b). PMF analysis showed that local emissions from vegetation are important
contributor for primary sugars (glucose, fructose and sucrose). Therefore, a drastic increase in
concentration of primary sugars in summer/autumn for P-II and P-III than P-I may be caused
by an increase in the emission of primary sugars by local vegetation under the influence of
meteorological conditions in the western North Pacific. However, a possible soil dust
contribution of primary sugar (sucrose) associated with an occasional air mass transport from
Southeast Asia cannot be excluded.

Similarly to primary sugars, a drastic increase in the concentrations of sugar alcohols was

observed for P-II and P-III compared to P-I period (Figure 8c). The concentrations of sugar
alcohols in P-II and P-III are 6 to 19 times higher than those of P-I in summer/autumn (Table
3; Figure 8c). Arabitol and mannitol are key sugar alcohols and reported as fungal and





microbial tracers, which contribute significantly to total SCs (Bauer et al., 2008b; Lewis and
Smith, 1967; Zhang et al., 2010). Microbes such as fungi and bacteria are significantly
increasing in the Asian and European countries (Yamaguchi et al., 2012). They are largely
transported towards downwind regions in the Pacific Ocean from the Asian Continent in
winter/spring under the influence of strong westerlies (Griffin et al., 2001, 2003, 2007; Hua et
al., 2007; Uno et al., 2009) and settled down by wet and dry deposition in the western North
Pacific according to air mass trajectories (Figure 3). Hirst and Stedman (1967) reported the life
time of the fungal spores, they studied long-range spores transport and discussed that the
fungus spores are between 2 and 200 μm in diameter (mostly 5 to 30 μ m); they have settling
velocity between 0.005 and 3.0 cm/sec with a mode of less than 1 cm/sec, and different fungal
spores vary significantly in shape and ornamentation. Most fungus spores are usually nearly
spherical and larger, have a specific gravity close to 1.0. Because of their larger mass, settle
velocity is at 1 to 40 cm/sec, with a mode of 3 cm/sec. This study suggested the possibilities of
the fungal spores long-range fungal spores.  Jeon et al. (2011) and Yamaguchi et al. (2012)
have analyzed aerosol samples collected during the Asian dust event over the Sea of Japan, and
identified similar groups of microbes (bacterial cells) transported from the source regions of
Asian dust. Consequently, bacteria and fungi grow extensively during summer/autumn, when
the climate conditions (i.e., higher RH and temperature) are favorable for their metabolic
activities (Morris et al., 2004). Accordingly, an increased transport of bioaerosols since the last
decade may have caused a drastic increase in the concentrations of sugar alcohols during P-II
and P-III compared to P-I period over the western North Pacific.
**3.4  Source apportionment of SCs**
To investigate the source apportionment of sugar components, the data sets of Chichijima
aerosols were subjected to positive matrix factorization (PMF) analysis. Based on the PMF
analysis, total five factors were determined to be significant to classify the sources of sugar



compounds (SCs). Five factors successfully explored the source profile for the individual sugar
component. Factor profiles resolved by PMF analysis are shown in Figures 6 and 9-11, where
percentages of each component summed for factors 1 to 5 are calculated to be 100%.

Vegetation factor (Figure 9) was dominated by xylose (75%), glucose (48%), and

fructose (36%). Xylose is significantly produced by gymnosperm and angiosperm (Cowie and
Hegdes, 1984; Sjostrom, 1981). Fructose and glucose are highly water-soluble sugar species,
and present in the bark and leaves of plants (Fu et al., 2012). Glucose is the second most
abundant sugar that contributed to this factor. Cowie and Hegdes (1984) reported higher
concentration of glucose in vascular plants and phytoplankton in the marine environment. The
SCs emitted by the vegetation during growing season significantly contribute to vegetation
factor. In Chichijima aerosols, glucose and fructose are significantly contributed in spring
(Figure 6b), summer (Figure 6c) and autumn (Figure 6d), Therefore, the respective factors in
Figure 6 are termed as a vegetation source for the both sugar species. This is reasonable
because plants started growing in spring and summer seasons. In autumn, leaf senescence and
decay result in the emission of glucose and fructose to the atmosphere.

Fungal and microbial factor (Figure 9) was characterized by trehalose (88%), mannitol

(64%), and arabitol (54%). These sugars that contribute to fungal and microbial factor are
associated with fungal spores, bacteria and yeast (Bauer et al., 2008a; Medeiros et al., 2006;
Wiemken, 1990). The three sugars are good tracers of fungal spores and microbes (Loos et al.,
1994; Rogge et al., 2007). Arabitol and mannitol are produced by a large variety of fungal
species (Ion et al., 2005; Medeiros et al., 2006), and considered as a suitable tracer for fungal
and bacterial metabolic activities (Bauer et al., 2008b; Elbein et al., 1974; Rogge et al., 2007).
Arabitol is strongly correlated with mannitol (r=0.88), suggesting similar sources for both
species (Table 2) (Elbert et al., 2007). Fungi, bacteria, and other microbes in soils are the main
sources for trehalose (Graham et al., 2003; Rogge et al., 2007; Simoneit et al., 2004). An



excellent correlation of trehalose with arabitol and mannitol suggested similar sources in the
marine environment (Table 2) (Lewis and Smith, 1967).
Sugar alcohols have been proposed as tracers for microbes and fungal spores (Bauer et
al., 2008b; Ion et al., 2005; Medeiros et al., 2006; Rogge et al., 2007). The fungal and
microbial activities are considered higher during summer and autumn due to higher
temperature and RH. The above discussions for the sources of arabitol, mannitol and trehalose
are well supported by the seasonal PMF analysis. Arabitol and mannitol are well contributed in
summer (Figure 6c, fungal factor and mixed factor) and autumn (Figure 6d, fungal and
vegetation factor). Correspondingly, trehalose also contributed in summer (Figure 6c, mixed
factor) and autumn (Figure 6d, microbial factor). Therefore, significant contributions of
arabitol, mannitol and trehalose are observed during the respective seasons in Chichijima
aerosols.
Mixed factor (Figure 9) is associated with erythritol (94%), arabitol (44%), mannitol
(34%), inositol (32%), glucose (24%), and fructose (31%). Due to the highly miscellany
characteristics of fungi, other microbes, and plant debris, it is quite difficult to specify the
particular source for individual sugar species (Percival, 1970). Arabitol and mannitol are also
attributed to the vegetation (photosynthesized by mature leaves) (Burshtein et al., 2011;
Pashynska et al., 2002). The contributions of arabitol and mannitol in winter (Figure 6a, mixed
factor) and spring (Figure 6b, mixed factor) indicate other sources than fungal spores in the
Chichijima aerosols. Because the fungal and microbial growth is less important in winter and
spring compared to summer/autumn. Therefore, this factor (Figure 9) should be associated with
mixed sources from microbial and vegetational activities. Contemplating of mixed sources is
very likely because sugar species that are highly apportioned to vegetation factor contribute, to
some extent, to the same sources that are responsible to fungal and microbial factor (Figure 9).
BB (biomass burning) factor (Figure 9) is loaded significantly with levoglucosan (94%),
mannosan (91%), and galactosan (90%), and moderately with xylose (20%). These species are





associated with biomass burning (Fraser and Lakshmanan, 2000; Graham et al., 2002;
Simoneit, 2002). Nolte et al. (2001) and Medeiros et al. (2006) also reported that biomass
burning-influenced aerosols are enriched with levoglucosan, mannosan, and galactosan.
Kawamura et al. (2003) and Mochida et al. (2010) reported that biomass-burning products are
abundantly transported to Chichijima under the influence of westerly winds during
winter/spring (Figure 3). These results are well supported by the fact that BB factor is
associated with SCs, which are derived from biomass burning in East Asia. The seasonal PMF
analysis (Figure 6) also supports the above explanation; BB products are contributed highest in
winter/spring (18.8% and 6.3%) under the influence of westerly winds, followed by
summer/autumn (3.8% and 4.1%) for total sugars in the aerosol samples collected from
Chichijima Island.

Pollen factor (Figure 9) is characterized by high loading of sucrose (91%), fructose

(35%), and inositol (33%); these sugar species are associated with airborne pollen sources.
Previous studies have also reported that sucrose is an excellent tracer for airborne pollen grains
of flowering plants (Pacini, 2000; Graham et al., 2003; Wang et al., 2008; Medeiros et al.,
2006). Fructose is well correlated with inositol (r=0.57), indicating a similar origin for both
sugar species (Table 2). Two prominent peaks of sucrose, fructose, and inositol, which
appeared in late winter to early spring or summer, also indicate a similar source of those sugars.
Contributions of sucrose (pollen factor) in winter (39.5%; Figure 6a) and spring (36%; Figure
6b) are supported the sources of airborne pollen for sucrose in Chichijima Island. The sucrose
contribution (soil dust factor) in non-flowering seasons, i.e., summer (17.1%; Figure 6c) and
autumn (8.4%; Figure 6d) indicates different sources for sucrose in Chichijima aerosols.
According to the seasonal PMF analysis (Figure 6), we termed additional sources of sucrose in
Chichijima Island as soil dust factor as well as pollen factor.

In Figure 6a, b, c and d, PMF analysis for seasonal source identification indicated

variable contributions of individual SCs in different seasons according to their seasonal source





origin. In winter (Figure 6a), airborne pollen (39.5%) contributed highest followed by
vegetation, microbial, fungal (mixed) (31.2%), biomass burning (18.8%) and microbial
(10.5%) sources. However, in spring, vegetation, microbial, and fungal (mixed) sources
(37.2%) contributed almost equal to the airborne pollen (36.0%) followed by vegetation
(21.2%) and biomass burning (6.3%) sources. The vegetation, microbial, fungal (mixed)
(54.2%), fungal (25.2%), soil particles (17.1%) and biomass burning (3.8%) sources are
characterized to maximize in summer season. Similar to summer, vegetation and fungal
(mixed) sources (71.2%) are also leading to the contribution of total SCs followed by microbial
(16.4%), soil particles (8.4%) and biomass burning (4.1%) in total SCs observed in autumn
season for Chichijima aerosols.

Overall, average contributions of each factor to measured SCs as resolved by the PMF

analyses are shown in Figure 10. Fungal and microbial factor accounts for 41% of total SCs
measured. The emission from microbes including fungal spores was found as a dominant
contributor to total SCs. Mixed factor (27%) indicates a common involvement of fungal,
microbial and vegetation sources. Figure 11 shows annual trends in % contributions of five
source factors to SCs in Chichijima aerosols. Fungal and microbial factor, and mixed factor
contributed higher than other sources for SCs during 2001 to 2013. However, no clear trends in
annual % contributions were observed for both source factors during thirteen-years study
period. The sugar species assigned as pollen tracers were found to contribute 18% in pollen
factor. Vegetation accounts for 11% of total SCs, indicating less emission from vegetation as
compared to fungi and microbes. As indicated by BB factor, biomass-burning source
contributes only 3% of total SCs.

Interestingly, we found an increasing trend in % contribution of vegetation, pollen, and

BB factors to SCs in 2006 to 2013 (see Figure 11). Sugar components, which are contributed
from pollen (sucrose, fructose, and inositol), vegetation (glucose and fructose), and biomass
burning sources (levoglucosan, galactosan, and mannosan), also show a similar increasing



trend for the period of 2006 to 2013 (Figure 7). The increased annual trends of BB and pollen
factors might be due to an enhanced long-range transport of airborne pollens and biomass
burning products from the Asian continent to the western North Pacific under the influence of
strong westerly winds. The increasing annual trends in % contribution of vegetation factor to
SCs may denote a significantly increased activity of local vegetation in Chichijima Island from
2006 to 2013, which could be involved with a recent global warming especially in the western
North Pacific region (http://climate.nasa.gov/vital-signs/global-temperature/).

**4  Summary and conclusions**
In this study we reported thirteen-years of temporal, seasonal and decadal observations on
sugar compounds (SCs) in the aerosol samples collected at Chichijima Island in the western
North Pacific, an outflow region of Asian aerosols. We observed the highest abundances of
total SCs and primary sugars in summer, while sugar alcohols are almost equally distributed
during summer and autumn. Thus, seasonal variations are well regulated by the atmospheric
circulation and meteorological parameters of Chichijima Island. The seasonal distributions of
arabitol, mannitol and trehalose are strongly influenced by long-range transport of microbe-,
fungi- and bacteria-associated bioaerosols and their metabolic activities under the influences of
westerly winds and other meteorological parameters (high RH and temperature) in
summer/autumn.
Seasonal variation of sucrose is controlled by locally emitted and long-range transported
pollen from East Asia to Chichijima during spring bloom periods. On the other hand, the
increased concentrations of sucrose and fructose in summer may be caused by the local activity
of vegetation and possibly by the atmospheric transport of plant root-associated soil dust
particles potentially delivered from East and Southeast Asia with the occasional transport of air
mass from the respective regions.



686   PMF analysis of long-term observations clearly indicated specific sources for individual

687 SCs during different seasons. The results separate biogenic emissions into two parts, i.e.,

688 vegetation and microbes including fungal species. The emissions from vegetation, pollen as

689 well as microbial activities contributed almost 97% of total SCs determined, with the

690 remaining fraction being derived from biomass burning activities. In the present decadal study,

691 we found a drastic increase in the concentrations of sugar alcohols and primary sugars during

692 2001-2003 and 2010-2013, which may be explained by a significant transport of bioaerosols in

693 last decades from East Asia to Chichijima Island in the western North Pacific.

695 **Acknowledgements**

696 We acknowledge the financial support by Japan Society for the Promotion of Science (JSPS)

697 through grant-in-aid Nos. 19204055 and 24221001. The authors gratefully appreciate the

698 NOAA Air Resources Laboratory (ARL) for the provision of the HYSPLIT transport and

699 dispersion model (http://www.ready.noaa.gov). The data for this paper are available upon

700 request to the corresponding author (Kimitaka Kawamura, kkawamura@isc.chubu.ac.jp).



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





**Figure Captions**
**Figure 1.** Geographical location of Chichijima Island (27°04'N; 142°13'E; 254 m, asl) in the
western North Pacific.
**Figure 2.** The monthly variation of the meteorological parameters over Chichijima Island
during 2001-2013 (The error bars denote the standard deviations).
**Figure 3.** The seasonal ten-day air mass backward trajectories over Chichijima for 2012
(winter: Dec-Feb, spring: Mar-May, summer: Jun-Aug, autumn: Sep-Nov). The trajectory
calculations were performed everyday starting at Chichijima Island.
**Figure 4.** Temporal plots for the concentrations (ng m$^{-3}$) of sugar compounds in Chichijima
aerosol samples collected for 2001-2013 in the western North Pacific.
**Figure 5.** Monthly mean concentrations (ng m$^{-3}$) of sugar compounds in aerosol samples from
Chichijima Island in the western North Pacific during 2001-2013.
**Figure 6.** Seasonal source contributions to sugar compounds from various sources based on
PMF analyses. (BB – biomass-burning; Mixed – vegetation, fungal and microbial sources).
**Figure 7**. Annual mean concentrations (ng m$^{-3}$) of sugar compounds in aerosol samples
collected from Chichijima Island in the western North Pacific during 2001-2013.
**Figure 8.** The seasonal concentrations of anhydrosugars (biomass burning tracers), primary
sugars and sugar alcohols measured in Chichijima aerosols during three periods, i.e., P-I
(1990-1993), P-II (2001-2003) and P-III (2010-2013).
**Figure 9.** PMF analyses of sugar compounds in Chichijima aerosols based on the 2001-2013
data set. (BB – biomass-burning; Mixed – vegetation, fungal and microbial sources).
**Figure 10.** Source contributions to sugar compounds from various sources based on PMF
analyses. (BB – biomass-burning; Mixed – vegetation, fungal and microbial sources).
**Figure 11.** Annual trends in % contributions of five source factors: (a) vegetation, (b) fungal
and microbial, (c) mixed, (d) biomass burning (BB), and (e) pollen factors to SCs in
Chichijima aerosols. The data of 2005 are not plotted due to limited data points.













**Figure 1.**

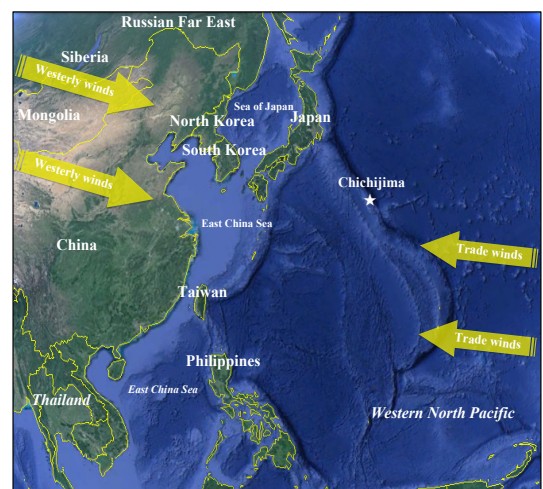




























**Figure 2.**

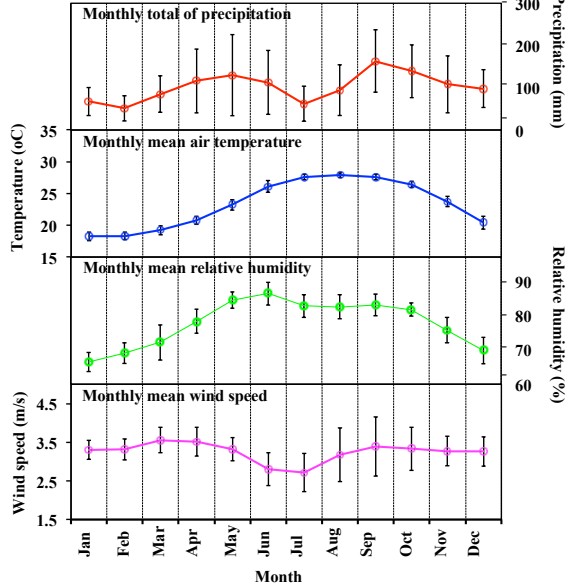




















**Figure 3.**

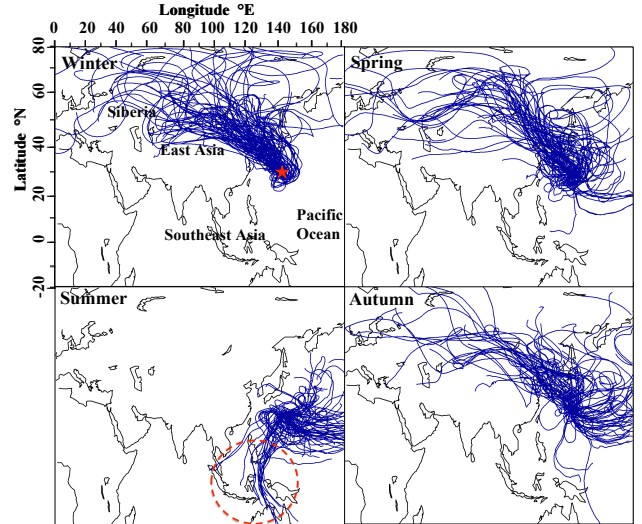




























**Figure 4.**

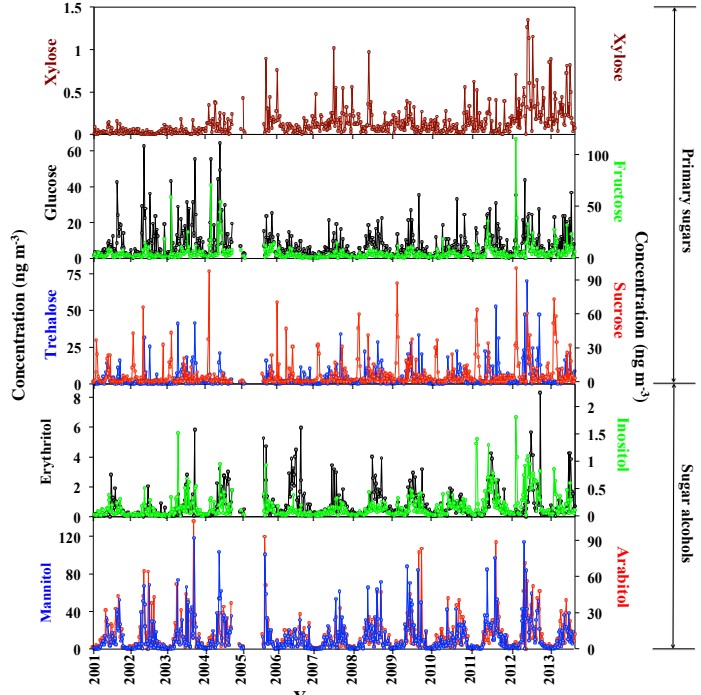


















**Figure 5.**

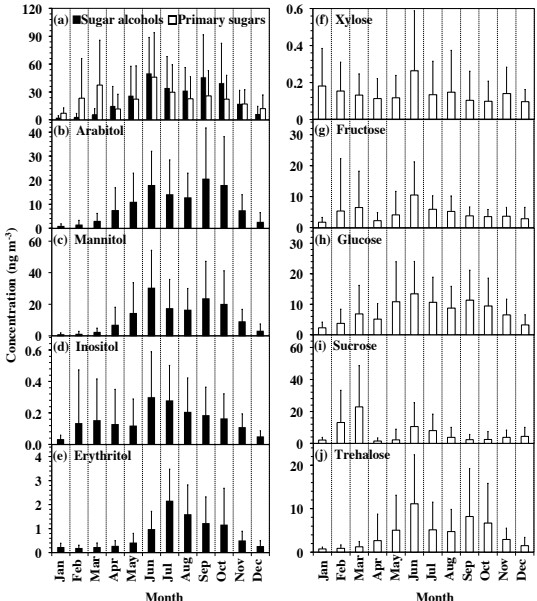























**Figure 6.**

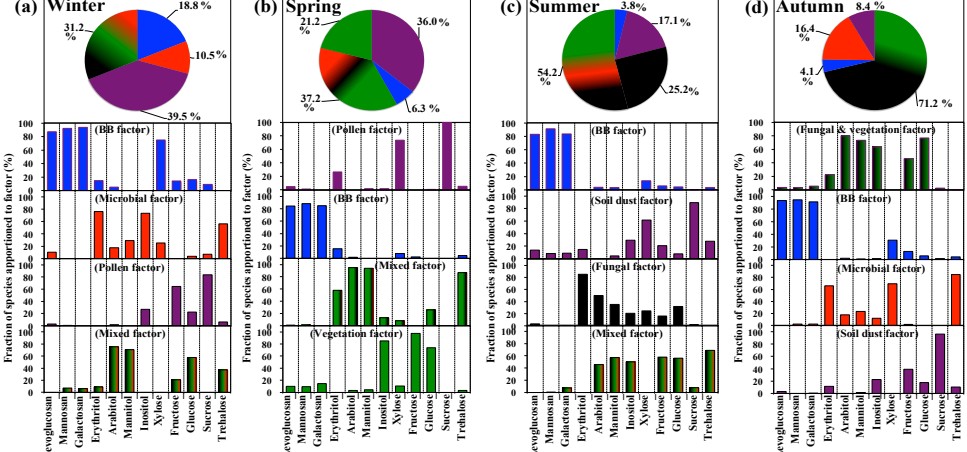



























**Figure 7.**

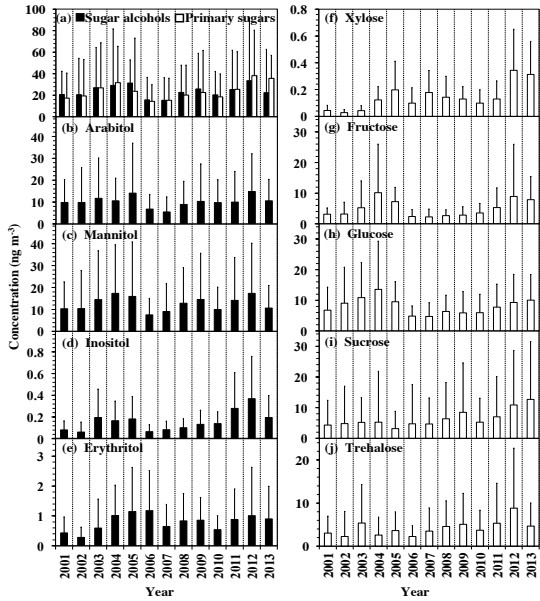

























**Figure 8.**

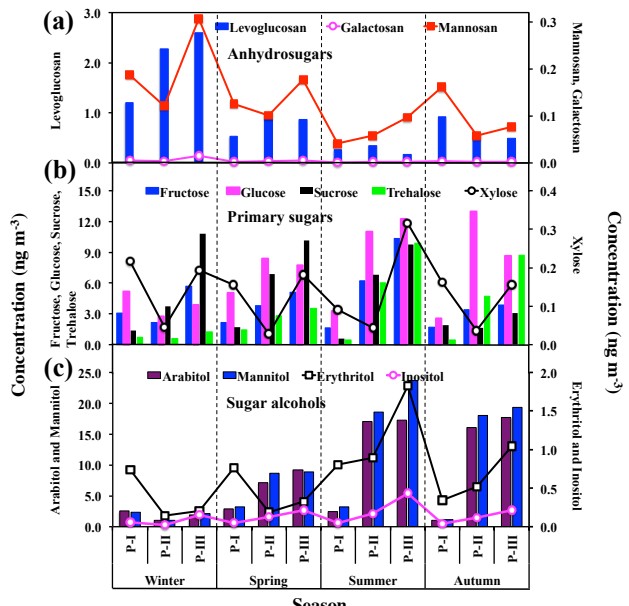




























**Figure 9.**

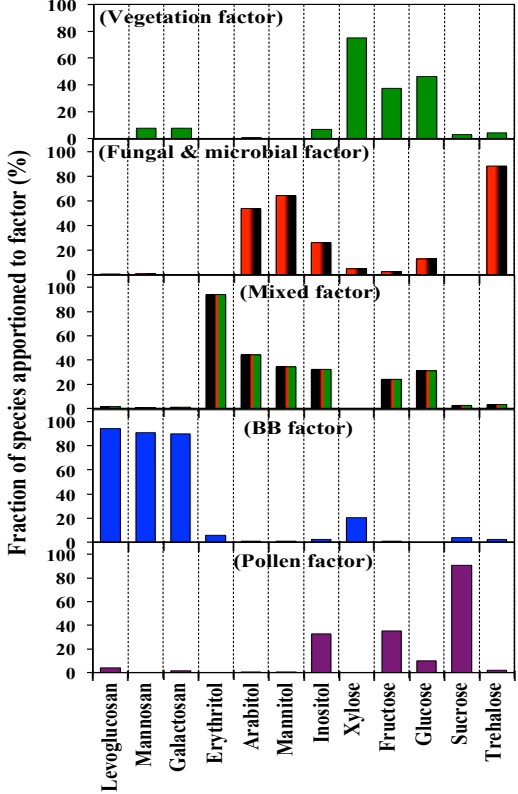

























**Figure 10.**

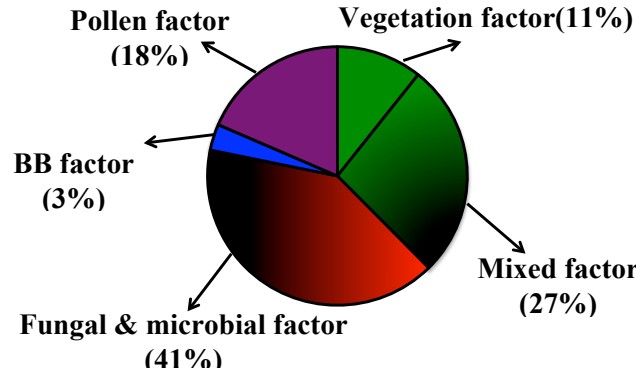





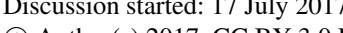











**Figure 11.**

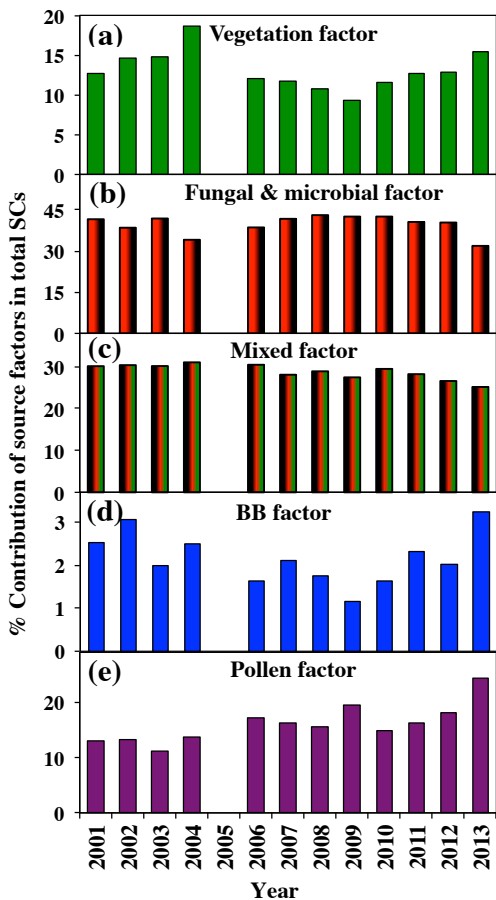






**Table 1.** Seasonal concentrations (ng m⁻³) of sugar compounds (SCs) in the aerosol samples collected at Chichijima Island in the western North Pacific during 2001-2013.

| Sugars | Winter[a] (n=139) | | | Spring[b] (n=155) | | | Summer[c] (n=146) | | | Autumn[d] (n=150) | | | 2001-2013 (n=590) | | |
|---|---|---|---|---|---|---|---|---|---|---|---|---|---|---|---|
| | Range | Mean±SD | Med[e] | Range | Mean±SD | Med[e] | Range | Mean±SD | Med[e] | Range | Mean±SD | Med[e] | Range | Mean±SD | Med[e] |
| **Primary sugars** | | | | | | | | | | | | | | | |
| Xylose | 0.05-0.89 | 0.14±0.15 | 0.12 | 0.001-0.56 | 0.12±0.11 | 0.08 | 0.002-1.35 | 0.18±0.26 | 0.08 | 0.004-0.89 | 0.11±0.14 | 0.07 | 0.001-1.35 | 0.14±0.18 | 0.08 |
| Fructose | 0.17-115 | 3.36±10.2 | 1.38 | 0.03-70.4 | 4.51±9.21 | 1.90 | 0.24-54.0 | 7.25±7.63 | 5.23 | 0.22-13.0 | 3.70±2.68 | 2.82 | 0.03-115 | 4.69±8.04 | 2.53 |
| Glucose | 0.27-23.3 | 3.11±3.53 | 2.13 | 0.05-62.6 | 7.68±10.3 | 4.36 | 0.23-64.3 | 11.0±9.02 | 7.99 | 0.72-55.5 | 9.25±8.63 | 5.77 | 0.05-64.3 | 7.79±8.80 | 4.73 |
| Sucrose | 0.02-73.4 | 6.60±13.1 | 2.08 | 0.005-100 | 8.80±18.0 | 1.30 | 0.003-66.0 | 7.31±11.5 | 2.58 | 0.002-31.1 | 2.76±4.35 | 1.14 | 0.002-100 | 6.43±12.9 | 1.71 |
| Trehalose | 0.03-10.5 | 1.03±1.26 | 0.72 | 0.006-47.2 | 2.93±6.08 | 1.23 | 0.03-70.2 | 7.06±8.49 | 4.46 | 0.04-52.5 | 6.09±8.81 | 2.46 | 0.01-70.2 | 4.30±7.28 | 1.61 |
| Σ Primary sugars | 0.49-223 | 14.2±28.2 | 10.6 | 0.09-281 | 24.2±43.8 | 18.2 | 0.51-256 | 32.8±36.9 | 50.7 | 0.99-153 | 22.0±24.6 | 31.1 | 0.28-176 | 23.3±25.7 | 30.9 |
| **Sugar alcohols** | | | | | | | | | | | | | | | |
| Erythritol | 0.03-1.17 | 0.23±0.18 | 0.18 | 0.008-2.25 | 0.31±0.28 | 0.23 | 0.07-5.70 | 1.55±1.21 | 1.18 | 0.05-8.32 | 0.99±1.16 | 0.56 | 0.01-8.32 | 0.77±1.01 | 0.37 |
| Arabitol | 0.12-21.2 | 1.73±2.60 | 0.96 | 0.04-70.8 | 7.13±9.50 | 4.44 | 0.24-64.5 | 15.1±12.9 | 12.0 | 0.57-106 | 15.8±18.3 | 8.83 | 0.04-106 | 9.99±13.6 | 4.97 |
| Mannitol | 0.10-23.9 | 1.89±2.81 | 1.11 | 0.16-114 | 7.95±13.8 | 4.07 | 0.25-104 | 21.7±19.7 | 16.9 | 0.59-118 | 18.2±19.9 | 9.15 | 0.10-118 | 12.5±17.5 | 5.54 |
| Inositol | 0.01-1.81 | 0.07±0.20 | 0.03 | 0.008-1.51 | 0.13±0.22 | 0.06 | 0.01-1.29 | 0.26±0.25 | 0.17 | 0.01-0.93 | 0.16±0.15 | 0.10 | 0.01-1.81 | 0.16±0.22 | 0.08 |
| Σ Sugar alcohols | 0.26-48.2 | 3.93±5.79 | | 0.22-188 | 15.5±23.8 | | 0.56-175 | 38.6±34.1 | | 1.22-234 | 35.1±39.5 | | 0.37-231 | 23.4±30.8 | |
| Σ SCs | 0.75-272 | 18.2±34.0 | | 0.31-469 | 39.8±67.6 | | 1.07-431 | 71.5±70.9 | | 2.21-387 | 57.0±64.2 | | 1.23-339 | 46.7±49.5 | |

[a] Winter (December-February), [b]Spring (March-May), [c]Summer (June-August) and [d]Autumn (September-November); Med.=Median



**Table 2.** Pearson correlation coefficients (r) for the dataset of sugars in Chichijima aerosols during 2001-2013 (n = 590).

| | Levoglucosan[a] | Mannosan[a] | Galactosan[a] | Erythritol | Arabitol | Mannitol | Inositol | Xylose | Fructose | Glucose | Sucrose | Trehalose |
|---|---|---|---|---|---|---|---|---|---|---|---|---|
| Levoglucosan | 1.00 | | | | | | | | | | | |
| Mannosan | 0.79 | 1.00 | | | | | | | | | | |
| Galactosan | 0.55 | 0.58 | 1.00 | | | | | | | | | |
| Erythritol | -0.18 | -0.12 | -0.16 | 1.00 | | | | | | | | |
| Arabitol | -0.16 | -0.09 | -0.16 | 0.48 | 1.00 | | | | | | | |
| Mannitol | -0.18 | -0.12 | -0.17 | 0.49 | 0.88 | 1.00 | | | | | | |
| Inositol | -0.06 | -0.02 | 0.10 | 0.35 | 0.49 | 0.58 | 1.00 | | | | | |
| Xylose | 0.20 | 0.32 | 0.34 | 0.15 | 0.18 | 0.23 | 0.42 | 1.00 | | | | |
| Fructose | 0.02 | 0.08 | 0.26 | 0.17 | 0.16 | 0.28 | 0.57 | 0.31 | 1.00 | | | |
| Glucose | -0.11 | 0.00 | -0.06 | 0.32 | 0.63 | 0.72 | 0.53 | 0.19 | 0.57 | 1.00 | | |
| Sucrose | 0.05 | 0.07 | 0.18 | -0.02 | -0.06 | 0.01 | 0.40 | 0.26 | 0.30 | 0.14 | 1.00 | |
| Trehalose | -0.10 | -0.05 | -0.10 | 0.33 | 0.73 | 0.80 | 0.55 | 0.33 | 0.22 | 0.54 | 0.13 | 1.00 |

[a] data from Verma et al. (2015)





**Table 3.** Comparisons of seasonal concentrations (ng m$^{-3}$) of primary sugars and relative contributions (%) of sugar compounds (SCs) in total SCs in Chichijima aerosols among 1990-1993[a], 2001-2003 and 2010-2013.

| Season | Anhydrosugars | | | Sugar alcohols | | | Primary sugars | | | Total Sugars | | |
|---|---|---|---|---|---|---|---|---|---|---|---|---|
| | 1990-93[a] | 2001-03[b] | 2010-13[b] | 1990-93[a] | 2001-03 | 2010-13 | 1990-93[a] | 2001-03 | 2010-13 | 1990-93[a] | 2001-03 | 2010-13 |
| Winter | 1.44 | 2.44 | 3.05 | 5.64 | 2.27 | 4.37 | 10.6 | 9.34 | 21.8 | 17.7 | 14.0 | 29.2 |
| % | 11.4 | 23.7 | 16.6 | 31.6 | 16.5 | 19.8 | 57.0 | 59.9 | 63.6 | | | |
| Spring | 0.68 | 1.12 | 1.10 | 6.91 | 16.2 | 18.6 | 10.5 | 21.3 | 26.7 | 18.1 | 38.6 | 46.4 |
| % | 5.08 | 5.17 | 4.04 | 39.2 | 39.7 | 42.0 | 55.7 | 55.1 | 53.9 | | | |
| Summer | 0.31 | 0.43 | 0.29 | 6.50 | 36.7 | 43.1 | 6.07 | 30.0 | 42.5 | 12.9 | 67.1 | 86.0 |
| % | 2.92 | 0.91 | 0.43 | 55.6 | 51.3 | 51.1 | 41.4 | 47.8 | 48.5 | | | |
| Autumn | 1.12 | 0.65 | 0.60 | 2.57 | 34.7 | 38.3 | 6.79 | 22.6 | 24.5 | 10.5 | 57.9 | 63.4 |
| % | 12.9 | 3.45 | 1.73 | 29.2 | 50.4 | 57.2 | 57.9 | 46.2 | 41.1 | | | |

[a] data from Chen et al. (2013), [b] data from Verma et al. (2015).