# Peer review of "Thirteen-years of observations on primary sugars and sugar alcohols over remote Chichijima Island in the western North Pacific"

_Atmospheric Chemistry and Physics, 2017_

## Referee Comment (RC1) · Anonymous Referee #2 · 12 Sep 2017

General

This is an interesting time-series paper on aerosol particle measurements of sugars and sugar alcohols at the remote island of Chichijima in the western north Pacific.

The authors should think about the numerical formats of their results. Sometime too many digits are listed which are uncertain in view of the calculated error width.

The English language of the manuscripts should be checked again.

Overall, I think this is an interesting paper for ACP publishable after minor revision.

[Figure]

Details

Title: I would suggest 'Thrteen years ...' without hypenation.

Abstract: The only numbers given here are the range mass concentrations of 'total sugar compound' . It would be great if more quantitative information could be given in the abstract, e.g. on the observed trends.

Line 64: For these contribution fractions only Simoneit et al. (2004) is referenced - are there any more recent fidnings on this which coul dbe referenced ? Can a range be given based on more references ?

Line 183 ff: I feel the section on PMF can considerably be shortened, much of this can be referenced.

Line 497: The sentence starting 'Thirteen-year...' can be left out.

Line 669, Summary and conclusion: Please make this consistent with the abstract. What is, finally, the conclusion of the paper ?

---

## Referee Comment (RC2) · Anonymous Referee #1 · 15 Sep 2017

General:

In general this paper reports valuable measurements for a number of sugars and sugar alcohols over a remote site in the middle of the open ocean over a long time period and conducts rational discussions for their seasonal and temporal trends, variation due to transportation and meteorological parameters, and source apportionment. The chemical, statistical, and other supplementary techniques for sample and data analyses are handled quite properly in order to provide and support their observations and conclusions.

[Figure]

As another anonymous referee mentioned, the English writing of the manuscripts should be further carefully checked and improved.

Overall, I think the paper is of decent quality for ACP to accept for publication after minor revision.

Individual Questions:

1. Although this might be complex and not so feasible, is there a better way to clearly present the types or contributions of parameters (i.e., local emission, wind, oceanic emission, etc) that dominate or influence the observed concentrations in different seasons?

2. Could you use the different wind directions, based on the backward air trajectory analysis, to categorize different sources of incoming aerosols, and investigate their influence? For example, westerly wind for aerosol transported from Asian continent; trade winds in summer/autumn for Pacific Ocean; no wind for local emission, etc..

Detailed Comments:

Line 73 - the sentence starting with "Previous studies..." needs revision in term of grammar.

Line 77 - the word "form" seems to be "from".

Line 82 - the sentence starting with "Thus, ..." can be removed, and the narrative in this and the next paragraph can be largely refined.

Line 183 - this section about PMF can be considerably shortened by removing unrelated content and using reference.

Line 510 - "20004a" seems to be "2004a"?

Line 514 - be consistent about using hyphenation, for example, "BB tracers" or "BB-tracers".

---

## Author Comment (AC1) · 13 Nov 2017

To
Prof. Jason Surratt
Editor
Atmospheric Chemistry and Physics (ACP)

**RE:** Submission of revised manuscript entitle "**Thirteen years of observations on primary sugars and sugar alcohols over remote Chichijima Island in the western North Pacific**" for publication in Atmospheric Chemistry and Physics

**Reference:** MS No.: acp-2017-480

Dear Prof. Surratt,

Thanks for the decision letter on our manuscript. We have revised the manuscript meticulously following the referees' comments and suggestions. The authors greatly appreciate the critical and useful comments by anonymous referee's that significantly improved the quality of our manuscript. Our responses and changes are included in the response letter as blue color. The changes in the revised manuscript are highlighted in yellow. Please find the response letter and revised manuscript.

We believe that revised manuscript is acceptable for publication in ACP.

Thanking you.

Sincerely yours,

Kimitaka Kawamura
Professor of Chemistry,
Chubu Institute for Advanced Studies,
Chubu University, Kasugai 487-8501,
Japan
E-mail: kkawamura@isc.chubu.ac.jp

**Authors' responses to the comments Referee # 1**

**General:**

In general this paper reports valuable measurements for a number of sugars and sugar alcohols over a remote site in the middle of the open ocean over a long time period and conducts rational discussions for their seasonal and temporal trends, variation due to transportation and meteorological parameters, and source apportionment. The chemical, statistical, and other supplementary techniques for sample and data analyses are handled quite properly in order to provide and support their observations and conclusions.
As another anonymous referee mentioned, the English writing of the manuscripts should be further carefully checked and improved.
Overall, I think the paper is of decent quality for ACP to accept for publication after minor revision.
Authors are thankful to the reviewer for the positive recommendation. We carefully checked and improved English writing of through out the manuscripts.

**Individual Questions:**

1. Although this might be complex and not so feasible, is there a better way to clearly present the types or contributions of parameters (i.e., local emission, wind, oceanic emission, etc) that dominate or influence the observed concentrations in different seasons?
**Response:** Authors are thankful to the reviewer for suggestion. We studied the atmospheric conditions over Chichijima Island and found that the aerosol composition of the Chichijima Island is strongly influenced by the atmospheric circulation in the western North Pacific with first westerly winds and second trade winds. The westerly winds transport continental air mass from East Asia during winter/spring, whereas the trade winds transport clean air mass from the central Pacific Ocean during summer/autumn. In our study we have discussed both local sources and long-range transport of SCs as follows:

1. The seasonal variation of sucrose in Chichijima aerosols was suggested with multiple sources, including both long-range transported and local emission. "The seasonal mean concentrations of sucrose are almost equal during spring (8.80±18.0 ng m-3), summer (7.31±11.5 ng m-3) and winter (6.60±13.1 ng m-3), except for autumn (2.76±4.35 ng m-3) (Table 1). The similar seasonal distributions suggest multiple sources of sucrose in Chichijima aerosols." The locally emitted pollen by flowering plant influenced the springtime sucrose concentration. The long-range transport of pollen from East Asia, Mongolia and Russian For East under the influence of westerly winds significantly contributes to the sucrose concentrations in the aerosols collected over Chichijima Island. (Please see lines 360-363 in the revised MS).

2. The contributions of arabitol and mannitol in the Chichijima aerosols were locally originated by the fungal and microbial metabolic activities under the favorable meteorological conditions (RH and temperature). "Consequently, bacteria and fungi associated with bioaerosols grow extensively during summer/autumn, when the climate conditions (i.e., higher RH and temperature) are favorable for their metabolic activities (Morris et al., 2004)." (Please see lines 532-534 in the revised MS).

3. The results of decadal study suggested an increased long-range air mass transport of bioaerosols from East Asia to the western North Pacific for the last decade. "Accordingly, an increased transport of bioaerosols for the last decade may have caused a drastic increase in the concentrations of sugar alcohols during P-II and P-III compared to P-I period over the western North Pacific." (Please see lines 535-537 in the revised MS).

4. The trade winds transport clean oceanic air mass from the central Pacific Ocean over Chichijima during summer/autumn. The higher concentrations of arabitol, mannitol, glucose, trehalose and erythritol are observed in Chichijima aerosols during summer/autumn. Although we also tried to discuss the oceanic sources for those SCs from the Pacific Ocean but due to the lack of sufficient data set and recent literature, we could not clarify either those sources are authentic or not. "Several studies have described the occurrence of fungi in marine environment……………Therefore, due to the inadequate data set, we doubt the marine contribution of sugar alcohols (arabitol, mannitol) in Chichijima aerosols." (Please see lines 455-467 in the revised MS).

Hence, we concentrated on local emission sources for those SCs, which showed higher concentrations during the trade wind periods (summer/autumn) over Chichijima Island. SCs are emitted from the local vegetation and metabolic activities of microbial and fungal species under favorable meteorological conditions.

2. Could you use the different wind directions, based on the backward air trajectory analysis, to categorize different sources of incoming aerosols, and investigate their influence? For example, westerly wind for aerosol transported from Asian continent; trade winds in summer/autumn for Pacific Ocean; no wind for local emission, etc.

**Response:** Thank you for the suggestions. We discussed a significant role of the atmospheric circulation (westerly and trade winds) in the seasonal concentrations of SCs in aerosol samples collected at Chichijima Island. We also discussed local sources for some SCs.

**Westerly winds:**
1. Please see lines 318-325 in the revised MS: "In winter/spring, Chichijima is influenced by strong westerly winds that deliver the air masses from the Asian continent including Mongolia, Russian Far East and North China, where vegetations are active. Consequently, declined concentrations of glucose in winter mean a depressed transport of glucose associated with continental bioaerosols from Asia despite long-range transport of Asian dusts due to strong westerly winds. The local vegetation (vascular plants) in Chichijima Island might be responsible to enhanced concentrations of glucose during growing season (spring and summer) and decaying periods of plant leaves (autumn)."

2. Please see lines 374-379 in the revised MS: "However, the possibilities of pollen transport from East Asia to Chichijima cannot be excluded because pollens can travel long distances with springtime high-speed winds by westerlies (Rousseau et al., 2006). The pollen grains emitted from flowering boreal forest in China, Mongolia, Siberia and Russian Far East, could significantly be delivered to the western North Pacific during spring, which may result in the contribution of sucrose and fructose to Chichijima aerosols."

3. Please see lines 387-402 in the revised MS: "These observations may support that westerly winds have delivered pollen grains from the Asian…………….which leads to smaller particles with longer residence time in the atmosphere."

4. Please see lines 528-537 in the revised MS: "This study suggested the possibilities of the long-range transport of the fungal…………….an increased transport of bioaerosols for the last decade may have caused a drastic increase in the concentrations of sugar alcohols during P-II and P-III compared to P-I period over the western North Pacific."

**Trade winds:**
Chichijima Island is dominated by trade winds during summer/autumn. We tried to find out some oceanic sources (like marine fungi) of SCs, transported under the influence of trade winds from the central Pacific Ocean but we did not find strong supportive data set to conclude the oceanic emissions of SCs.

1. Please see lines 455-467 in the revised MS: "Several studies have described the occurrence of fungi in marine environment (Jones, 1976; Kohlmeyer and Kohlmeyer, 1991; Moss, 1986). The fungal species eject spores…………….Therefore, due to the inadequate data set, we doubt the marine contribution of sugar alcohols (arabitol, mannitol) to Chichijima aerosols."

2. Please see line no. 405-415 in the revised MS: "In China and India there are two seasons (spring and winter) for wheat crops; winter wheat is harvested from mid-May to……………suggest the transport of sucrose associated with soil particles under the influence of occasional air mass transport in summer from Southeast Asia in summer (Figure 3)".

**Local Sources:**
The local sources are discussed for those SCs, which are dominant during summer/autumn.

1. Please see lines 334-337 in the revised MS: "Monthly mean concentrations of fructose show two prominent peaks in February-March and June-July, the latter peak may be due to the local vegetation in Chichijima (Figure 5g). The fructose peak in February-March may be influenced by air borne pollen grains in the spring bloom of flowering plants."

2. Please see lines 449-454 in the revised MS: "The meteorological factors such as RH and temperature significantly affect fungal and bacterial activity (Kim and Xiao, 2005; Malik and Singh, 2004). Higher RH and temperature are crucial in increasing fungal and bacterial growth (Sharma and Razak, 2003). Their maximum growth was observed under the condition of 92-100% RH (Ibrahim et al., 2011). Higher concentrations of arabitol and mannitol in summer and autumn may be caused by the increased fungal and bacterial activities in Chichijima Island."

3. Please see lines 471-477 in the revised MS: "The thirteen-year precipitation record over Chichijima Island shows that precipitations were lowered in July and August (Figure 2). The lower precipitation amount decreases the RH (Figure 2) and thus depresses the fungal and microbial activities. The lower precipitation also suppresses the moisture contents in the surface soil of Chichijima, which should result in a significant decline of local fungal and other microbial activities on the ground of Chichijima Island. Decreased precipitation might be a possible reason for the lower concentrations of arabitol, mannitol, and trehalose in July and August."

4. Please see lines 507-511 in the revised MS: "PMF analysis showed that local emissions from vegetation are important contributor for primary sugars (glucose, fructose and sucrose). Therefore, a drastic increase in the concentrations of primary sugars in summer/autumn for P-II and P-III than P-I may be caused by an increased emission of primary sugars by local vegetation under the influence of meteorological conditions in the western North Pacific."

5. Please see lines 552-556 in the revised MS: "In Chichijima aerosols, glucose and fructose are significant contributors in spring (Figure 6b), summer (Figure 6c) and autumn (Figure 6d), Therefore, the respective factors in Figure 6 are termed as a vegetation source for both sugar species. This is reasonable because plants started growing in spring and summer seasons. In autumn, leaf senescence and decay result in the emission of glucose and fructose to the atmosphere."

**Detailed Comments:**

**Line 73-** the sentence starting with "Previous studies..." needs revision in term of grammar.
**Response:** Corrected. "Sucrose is dominant sugar component in airborne pollen grains and plays a significant role in plant blossoming activity (Bieleski, 1995; Fu et al., 2012; Pacini, 2000)". (Please see lines 80-82 in the revised MS).

**Line 77-** the word "form" seems to be "from".
**Response:** Corrected. Please see line 84 in the revised MS.

**Line 82-** the sentence starting with "Thus, ..." can be removed, and the narrative in this and the next paragraph can be largely refined.
**Response:** The sentence was removed and the next paragraph has been modified. Please see lines 89-100.

**Line 183-** this section about PMF can be considerably shortened by removing unrelated content and using reference.
**Response:** According to referee's suggestions, "Section 2.4 Positive matrix factorization (PMF) analysis" was significantly shortened. We added a new sentence: "The detailed discussions of the determination and application of the PMF are reported in Norris et al. (2008), Paatero et al. (2002) and Zhou et al. (2004)." (Please see lines 206-208 in the revised MS).

**Line 510-** "20004a" seems to be "2004a"?
**Response:** Corrected. (Please see line 495 in the revised MS).

**Line 514-** be consistent about using hyphenation, for example, "BB tracers" or "BB-tracers".
**Response:** We decided to use "BB-tracers" throughout the manuscript. (Please see the revised MS).

[revised manuscript text omitted]

[a] data from Chen et al. (2013), [b] data from Verma et al. (2015).

---

## Author Comment (AC2) · 13 Nov 2017

To
Prof. Jason Surratt
Editor
Atmospheric Chemistry and Physics (ACP)

**RE:** Submission of revised manuscript entitle "**Thirteen years of observations on primary sugars and sugar alcohols over remote Chichijima Island in the western North Pacific**" for publication in Atmospheric Chemistry and Physics

**Reference:** MS No.: acp-2017-480

Dear Prof. Surratt,

   Thanks for the decision letter on our manuscript. We have revised the manuscript meticulously following the referees' comments and suggestions. The authors greatly appreciate the critical and useful comments by anonymous referee's that significantly improved the quality of our manuscript. Our responses and changes are included in the response letter as blue color. The changes in the revised manuscript are highlighted in yellow. Please find the response letter and revised manuscript.

   We believe that revised manuscript is acceptable for publication in ACP.

Thanking you.

Sincerely yours,

Kimitaka Kawamura
Professor of Chemistry,
Chubu Institute for Advanced Studies,
Chubu University, Kasugai 487-8501,
Japan
E-mail: kkawamura@isc.chubu.ac.jp

**Authors' responses to the comments Referee # 2**

**General**

This is an interesting time-series paper on aerosol particle measurements of sugars and sugar alcohols at the remote island of Chichijima in the western north Pacific.

The authors should think about the numerical formats of their results. Sometime too many digits are listed which are uncertain in view of the calculated error width.

The English language of the manuscripts should be checked again.

Overall, I think this is an interesting paper for ACP publishable after minor revision

Authors are thankful to the reviewer for the positive recommendation. We carefully checked and improved English writing of through out the manuscripts.

**Details**

**Title:** I would suggest 'Thrteen years ...' without hypenation.
Response: We deleted the hypenation from Thrteen years… in the title. (Please see the title in the revised MS).

**Abstract:** The only numbers given here are the range mass concentrations of 'total sugar compound'. It would be great if more quantitative information could be given in the abstract, e.g. on the observed trends.
Response: According to the reviewer's suggestions, we added some more information in the abstract. (Please see lines 34-36 in the revised MS).

**Line 64:** For these contribution fractions only Simoneit et al. (2004) is referenced - are there any more recent findings on this, which could be referenced? Can a range be given based on more references?
**Response:** According to the reviewer's suggestions, we modified the sentences and added new phrase with additional references. (Please see lines 65-73 in the revised MS.)
"Sugar compounds (SCs) contribute 13–26% and 63% of the total compound mass in the continental and marine aerosol samples, respectively (Simoneit et al., 2004a; 2004b). Yttri et al. (2007) reported that sugars (fructose, glucose, sucrose, trehalose) accounted for 0.6-3.1% of WSOC at urban and suburban sites in Norway. Tominaga et al., (2011) analyzed aerosol samples collected from urban and forest suburban sites from Japan and reported that sugars (arabinose, fructose, galactose, glucose, mannose, rhamnose, and xylose) accounted for 2.1% and 4.5% of WSOC in the fine and coarse mode ranges at Yokohama, respectively, and for 3.0% and 7.2% at Mt. Oyama, respectively."

We added a new reference: Tominaga, S., K. Matsumoto, N. Kaneyasu, A. Shigihara, K. Katono, N. Igawa (2011), Measurements of particulate sugars at urban and forested suburban sites, Atmos. Environ. 45, 2335-2339.

**Line 183:** I feel the section on PMF can considerably be shortened; much of this can be referenced.
**Response:** According to referee's suggestion, "Section 2.4 Positive matrix factorization (PMF) analysis" was significantly shortened. We added a new sentence: "The detailed discussions of the determination and application of the PMF are reported in Norris et al. (2008), Paatero et al. (2002) and Zhou et al. (2004)." (Please see lines 206-208 in the revised MS.)

**Line 497:** The sentence starting 'Thirteen-year...' can be left out.
**Response:** According to referee's suggestions, we deleted the sentences starting from thirteen-years …..). (Please see line 484 in the revised MS).

**Line 669**, Summary and conclusion: Please make this consistent with the abstract. What is, finally, the conclusion of the paper?
**Response:** According to referee's suggestions, we modified the abstract and conclusion section as follows.

We modified sentence: "The primary sugars (glucose and fructose) maximized in summer, possibly due to an increased emission of vegetation products from the local vascular plants in Chichijima." (Please see lines 27-29 in the revised MS).

Added a new sentence: "Sucrose and trehalose were found to present increasing trends from 2001 to 2013, while total sugar components did not show any clear trends during thirteen year periods." (Please see lines 34-36 in the revised MS.)

We significantly modified summary and conclusion sections in the revised MS. (Please see lines 651-687).

[revised manuscript text omitted]

[a] data from Chen et al. (2013), [b] data from Verma et al. (2015).